# Modular Chitosan-Based Adsorbents for Tunable Uptake of Sulfate from Water

**DOI:** 10.3390/ijms21197130

**Published:** 2020-09-27

**Authors:** Bernd G. K. Steiger, Lee D. Wilson

**Affiliations:** Department of Chemistry, University of Saskatchewan, 110 Science Place, Saskatoon, SK S7N 5C9, Canada; bes241@mail.usask.ca

**Keywords:** adsorption, chitosan beads, sulfate, surface modification, calcium imbibing, cross-linking

## Abstract

The context of this study responds to the need for sorbent technology development to address the controlled removal of inorganic sulfate (SO_4_^2−^) from saline water and the promising potential of chitosan as a carrier system for organosulfates in pharmaceutical and nutraceutical applications. This study aims to address the controlled removal of sulfate using chitosan as a sustainable biopolymer platform, where a modular synthetic approach was used for chitosan bead preparation that displays tunable sulfate uptake. The beads were prepared via phase-inversion synthesis, followed by cross-linking with glutaraldehyde, and impregnation of Ca^2+^ ions. The sulfate adsorption properties of the beads were studied at pH 5 and variable sulfate levels (50–1000 ppm), where beads with low cross-linking showed moderate sulfate uptake (35 mg/g), while cross-linked beads imbibed with Ca^2+^ had greater sulfate adsorption (140 mg/g). Bead stability, adsorption properties, and the point-of-zero charge (PZC) from 6.5 to 6.8 were found to depend on the cross-linking ratio and the presence of Ca^2+^. The beads were regenerated over multiple adsorption-desorption cycles to demonstrate the favorable uptake properties and bead stability. This study contributes to the development of chitosan-based adsorbent technology via a modular materials design strategy for the controlled removal of sulfate. The results of this study are relevant to diverse pharmaceutical and nutraceutical applications that range from the controlled removal of dextran sulfate from water to the controlled release of chondroitin sulfate.

## 1. Introduction

Saline ground and surface water deposits pose a significant concern to global water and food security, especially in developing countries. Sulfate is an important oxyanion component in saline ground water and aquifers due to its high solubility and mobility. However, the role of sulfate is often underestimated due to its relatively low toxicity [1]. At elevated sulfate levels (ca. 250–500 mg/L and greater), adverse health effects include diarrhea, especially for children since they are more susceptible than adults [2,3]. Climate change and increased irrigation for food production may exacerbate the detrimental effects of water salinity [4,5], where projections forecast negative impacts on global water security [6]. In countries such as the USA and Canada, sulfate contamination is attributed to the erosion of naturally abundant magnesium sulfate and gypsum deposits. Further, sulfur emissions by industry can reach critical levels in wastewater that contribute to ecosystem disruption. Sulfate from geochemical sources have been related to elevated sulfate levels that can range from 6500 mg/L to 20,000 mg/L in dug-outs of Saskatchewan, Canada’s agricultural bread basket [7,8].

Currently, sulfate remediation does not generally employ adsorption-based methods, even though adsorbent technology can provide benefits in terms of cost, environmental and maintenance over other types of physical and chemical removal methods [9,10,11,12,13,14,15]. Adsorption-based removal of sulfate offers a relatively low-cost approach with variable efficiency and selectivity [16]. The merits of the adsorption process are generally limited by the availability of an adsorbent material with suitable properties. In contrast to petrochemical-based ion-exchange resins for efficient adsorptive removal, there is renewed interest in the development of sustainable biopolymer adsorbents using chitosan-based materials. Chitin is a natural biopolymer with a poly(β-(1-4)-*N*-acetyl-d-glucosamine) structure that is the second most abundant biopolymer after cellulose [16,17]. Chitosan is typically obtained from chitin via deacetylation pathways, where the structural similarity of cellulose and chitosan relate to differences in the chemical functionality at the C2-carbon. The presence of –NH_2_ instead of –OH groups result in variable chemical reactivity along with a unique solubility profile of chitosan in organic and mineral acids (except for sulfuric and phosphoric acid), along with the metal chelation properties of the glucosamine monomer units of chitosan [18]. The solubility of chitosan results from the protonation of its amine groups which can coordinate with anionic species or undergo ion-exchange with multivalent metal cations, further revealing its unique adsorption properties [19,20]. Pristine chitosan has some limitations in terms of its low accessible surface area, low mechanical stability, and only partial accessibility of its functional groups, due to its semi-crystalline structure [18,21]. To address these limitations, synthetic modification of chitosan via physical or chemical cross-linking have been reported [22,23,24]. Examples of chemical cross-linkers used to modify the physicochemical properties of chitosan include epichlorohydrin and glutaraldehyde [25,26]. The controlled removal of oxyanions, such as arsenates and vanadates, has been reported [27,28,29], along with organic dyes and organophosphates from wastewater [26,30,31,32,33]. Modified chitosan adsorbents have been employed for the removal of oxyanions with variable solubility profiles and ionic charge that include sulfate, phosphate, and nitrate. [34,35]. 

In contrast with reported work on phosphate and nitrate adsorption, fewer studies are available on the adsorption of sulfate (cf. Appendix A), in agreement with recent reviews that outline the adsorptive removal of other types of oxyanions [36,37,38,39]. The ubiquitous occurrence of sulfate in aquatic environments relate to its role as a competitive anion that influences the uptake efficiency of other target oxyanion species [40]. Sulfate as a competitor anion is supported by a recent study of mixed anions (Cl^−^, HCO_3_^−^, and SO_4_^2−^) [41]. Similar to chitosan, sparse reports are available for chitin materials in the form of shrimp shells and grafted composite materials have been reported for sulfate removal from aqueous sources [15]. The chitosan bead systems offer various benefits over its pristine form, e.g., reduced back pressure in column application and increased flow rate [42]. Therefore, the present work relates to the to the modular materials design of chitosan beads with tunable sulfate adsorption properties via a phase-inversion synthesis of chitosan beads with sequential glutaraldehyde cross-linking and bead surface modification by Ca^2+^ doping (cf. Figure 1). Through this modular materials design, it is posited that chitosan beads prepared by this route will display incremental sulfate adsorption properties, in a parallel fashion as reported for other metal-oxyanion (CrO_4_^2−^) systems with other co-adsorbed metal species [35,43,44,45,46]. Furthermore, this study builds upon work related to the adsorptive removal of oxyanions (phosphate and nitrate) outlined in a recent review [36]. This study exemplifies a sustainable and scalable synthesis that embodies elements of green chemistry for the design of chitosan-based adsorbents for sulfate as model anion for sulfate or sulfonated species (e.g., methyl orange, reactive black 5, dextran sulfate) [47]. It is posited that an improved understanding of the adsorption properties of sulfate, as a model anion, may relate to improved insight on the adsorption/desorption characteristics of relevant carrier systems for the uptake of lipid adducts or release of chondroitin sulfate. Other potential applications include dietary control and osteoarthritis strategies, respectively [48,49].

In light of the limited reports on sulfate adsorption, the present work was motivated to develop improved chitosan-based adsorbents for controlled sulfate removal due to the relevance of sulfate species to global water security and pharmaceutical/nutraceutical applications for this thematic journal issue. 

## 2. Results

As outlined above, the limited research on suitable biopolymer adsorbents for inorganic sulfate led us to the modular adsorbent design strategy in Figure 1. Following the modular synthesis of the chitosan bead systems herein, several complementary techniques are used to characterize the structure and physicochemical properties of the bead systems relevant to adsorption of sulfate, as outlined below.

### 2.1. ^13^C NMR Spectral Result

^13^C-solid state NMR spectroscopy is a sensitive method for detecting ^13^C nuclei in unique chemical environments. The resulting ^13^C NMR spectra (cf. Figure 2) reflect differences in chemical modification of chitosan due to differences in abundance and chemical shift information. Thus, ^13^C NMR spectral results enable evaluation of the role of different levels of chemical modification that result upon cross-linking with glutaraldehyde at variable levels and calcium imbibing, as compared with pristine chitosan.

The ^13^C solids NMR spectra reveal that imine cross-linkages are more evident at higher ratios (1:1 and above ca. 140 ppm for newly formed N=C-groups; see peak 11 in Figure 2C,D), along with signatures ca. 20 and 45 ppm (peak number 9 and 10) for the presence of aliphatic carbons of glutaraldehyde. This trend corresponds to the level of glutaraldehyde cross-linking, where a stoichiometric excess of glutaraldehyde is required for effective cross-linking at these conditions. Also, the spectral intensity of the N=C-groups in the bead are influenced by the presence of Ca^2+^ due to the role of chelation on the cross-polarization dynamics of the system [50]. Aside from the NMR spectral confirmation of the products reported in Figure 2, there is a need to characterize the thermal stability of the biopolymer network since thermal analysis was reported to be sensitive to be sensitive toward the structure and bonding variability of the chitosan composites, as described elsewhere [51]. The results in Figure 3 outline the thermal analysis profiles for the bead systems.

### 2.2. Thermogravimetric Analysis (TGA) Results

The TGA results obtained for the chitosan materials provide a complementary approach for studying the physical properties of structurally similar systems [51,52,53]. Herein, we compared the TGA profiles of the chitosan beads after incremental chemical modification (cross-linking and imbibing) to reveal the role of the modular synthesis (cf. Figure 1) on the structure and thermal stability of such systems.

In Figure 3, there are several observable thermal events that relate to the chemical nature of the chitosan bead system. The decomposition event for non-cross-linked chitosan beads can be described as a relatively sharp thermal event near 300 °C with a relatively narrow temperature interval that begins near 275 °C and spans up to ca. 325 °C. In comparison, the cross-linked chitosan beads have a lower average onset decomposition temperature that occurs near 250 °C that cover a broader range up to ca. 330 °C, including other thermal events near 450 °C. The lower temperature onset of cross-linked chitosan and the broadened temperature range reveal its overall lower thermal stability, as compared with beads without cross-linking. In a previous report, unmodified chitosan was found to have more crystalline domains due to efficient hydrogen bonding, as compared to cross-linked chitosan [23]. Cross-linking results in defects of the chitosan network similar to that of Ca^2+^ species imbibed within the bead matrix (relative to pristine chitosan) by creating defects within the hydrogen bond network of the biopolymer. A comparison of the imbibed and non-imbibed systems in Figure 3 reveals trends in the thermal profiles that further support the role of surface modification (cross-linking) and incorporation of calcium species from the modular bead synthesis (cf. Figure 1). The TGA results reveal variable thermal stability of the bead systems, while the NMR results show distinctive structural effects of the chitosan framework upon cross-linking and imbibing of Ca^2+^. The cross-linking of beads with glutaraldehyde affect the internal and surface structure of such biopolymer systems due to “pillaring effects” of the chitosan bead [54]. The role of covalent cross-linking of chitosan and surface modification with Ca^2+^ correlate with trends in the thermal stability of the bead systems, as noted in the TGA profiles. The incorporation of cross-linker resulted in secondary thermal events and the role of Ca^2+^ imbibing is observed in Figure 3, as noted by band broadening, and temperature shifts in the TGA profile. The latter trend shows parallel agreement with attenuated motional dynamics of modified chitosan according to the ^13^C NMR spectral broadening observed in Figure 2. Likewise, the TGA band broadening noted for the modified chitosan depend on the cross-linker content since the formation of imine linkages alter the H-bond network, as compared with pristine chitosan. 

### 2.3. Solvent Swelling in Water & Dye-Based Surface Area Estimates

A recent study of the solvent swelling properties was shown to provide complementary insight on the hydration and adsorption properties of chitosan biopolymers [55]. A simple yet informative method to measure solvent swelling can be estimated through the gravimetric determination of water uptake since chitosan-based adsorbents are known to undergo swelling in aqueous media [23]. In Figure 4, the results for the equilibrium solvent swelling in water are shown for chitosan beads with incremental structural modification (cross-linking and Ca^2+^ imbibing). While the non-modified beads show the highest level of water swelling (ca. 150%), the cross-linked beads show reduced swelling due to the role of incremental cross-linking since it is known to limit the volumetric expansion of such cross-linked materials. Calcium imbibed beads with a cross-linking ratio (CL-ratio) show variable swelling as follows: 97% (1:10), 85% (1:5), 58% (1:1), and 53% (5:1).

In general, chitosan beads with higher cross-linking ratios often possess greater rigidity with less flexibility and greater steric effects, as evidenced by the reduced accessibility of the polymer chains [54]. In contrast to unmodified chitosan beads without cross-linking, the bead systems with incremental glutaraldehyde cross-linking show reduced water uptake and swelling. This trend in variable hydration and steric effects is supported by another study that reports on the variable accessibility of N-containing functional groups of chitosan upon cross-linking [56]. The structural characterization of the different bead systems affirmed the effects of modification from the modular synthesis of chitosan in Figure 1. However, the role of chitosan modification and its relationship to the structure–adsorption properties of the bead systems with sulfate requires further study. Previous work indicates that cross-linking with glutaraldehyde converts the accessible amine groups to imine linkages and alteration of the bead physical properties [24,25,57,58], in contrast to pristine chitosan beads without synthetic modification. 

Gas adsorption is a conventional characterization method for estimating the surface area (SA) of adsorbent materials. In the case of biopolymer materials that undergo swelling in aqueous media, dye-based adsorption offers an alternative approach for estimating the adsorbent SA. The measurements were conducted at the isosbestic point of PNP to avoid errors due to changes in solution pH [59]. Herein, a dye-based method was employed, where *p*-nitrophenol (PNP) served as the dye probe for estimating the bead SA (cf. Table 1).


(1)$$\mathrm{SA}\left(\frac{m^{2}}{g}\right)=\frac{Q_{m}*N*\sigma_{\mathrm{PNP}}}{Y}$$


Q_m_ can be obtained from an analysis of the adsorption results using the Sips isotherm model (cf. Equation (2), where Q_m_ represents the maximum monolayer adsorption capacity at equilibrium. The other terms in Equation (1) are also defined: N is Avogadro’s number, σ is the cross-sectional area of the adsorbate, and Y is the coverage factor (*Y* = 1 for PNP).

The effects of non-uniform drying can result in beads with non-uniform shape and size for systems without cross-linking. The size estimates of the various beads along with the mean, median and standard deviation are listed in Appendix A. By contrast, cross-linked beads show more uniform spherical shape and size distributions that seem to stabilize the bead structure upon drying. Thus, the cross-linked bead systems show a general increase in SA over unmodified bead systems due to “pillaring effects” [54] related to cross-linking. While greater cross-linking results in reduced solvent swelling (cf. Figure 4), greater surface accessibility of the dye binding sites is supported by the greater SA estimates for beads with greater cross-linking in Table 1. This trend is contrasted to the unmodified (no cross-linking) bead systems. Notwithstanding the role of steric effects at elevated levels of cross-linking, incremental levels of glutaraldehyde contribute to a greater adsorption of PNP, in agreement with an independent study reported by Mohamed et al. [54] for the adsorption of naphthenate anion congeners by cross-linked chitosan polymers. 

### 2.4. X-ray Photoelectron Spectroscopy (XPS) Results

XPS results for unmodified and cross-linked beads, with and without calcium imbibing are shown in the Appendix A. The unmodified and non-imbibed cross-linked beads show a small calcium signature that likely originates from the deacetylation process from the as-received commercial product. Imbibing beads with calcium resulted in a more pronounced calcium signature that further support the role of cation coordination onto the bead surface. A small but appreciable nitrogen signature appeared after cross-linking with glutaraldehyde, where a peak shift occurs upon calcium imbibing that signifies a change in coordination environment due to the presence of glutaraldehyde.

### 2.5. Sulfate Adsorption Results 

Phase-inversion synthesis of chitosan beads with subsequent cross-linking and metal ion imbibing results in the incorporation of Ca^2+^ species onto the bead surface. Modular synthesis of this type was shown to result in enhanced uptake properties of organophosphates, as compared with unmodified bead systems [26]. By analogy, incremental sulfate adsorption properties are posited for materials prepared via the modular design approach (cf. Figure 1). The adsorption isotherm Sips modeling [60] results for sulfate were obtained by the turbidity method [61] The trends in the adsorption of sulfate uptake reveal how the modular synthesis of chitosan beads relate to the incremental variation in the sulfate adsorption properties (cf. Appendix A). In Figure 5, the adsorption isotherms illustrate trends in the sulfate adsorption that relate to the role of chemical modification of the bead systems at conditions relevant to ground water environments.

The adsorption isotherm results reveal that the beads efficiently remove sulfate from aqueous solution at levels below 1000 ppm, where the overall adsorption capacity scales with the initial sulfate concentration. Although the bead systems exhibit reduced swelling behavior with increased cross-linking, the uptake capacity increases with greater cross-linking, further highlighting the primary role of adsorption sites at the chitosan bead surface. This trend parallels that for the dye adsorption results of PNP (cf. Table 1). By contrast, the negligible role of mass transfer of sulfate into the bead core due to diffusion effects can be inferred from confocal microscopy results of anionic dyes with chitosan beads reported elsewhere [58]. Thus, the adsorption sites within the bead core for such systems is inferred to play a secondary role in sulfate adsorption. The uptake reached a maximum of ca. 140 mg/g with the 5:1 cross-linked and calcium-imbibed material, further supporting the key (primary) role of the bead surface adsorption sites for sulfate. This trend concurs with the formation of imine surface sites and their key role in anion uptake properties [62], in parallel agreement with the dye adsorption results presented in Table 1. Whereas the main contributions of the modified beads are concluded to occur mainly at the bead surface sites of the modified chitosan, secondary adsorption of sulfate in the chitosan bead core contribute less due to limited adsorbate diffusion. A similar binding mode for sulfate uptake is proposed, as reported elsewhere [63] for the adsorption of urea by Cu(II) imbibed form of chitosan (cf. Scheme 1 in [63]). The observed trend in sulfate uptake relates to glutaraldehyde cross-linking, as evidenced by the reduced bead swelling, since incremental cross-linking is inferred to contribute to reduced bead expansion. The incremental formation of imine linkages on the bead surface favor anion adsorption due to the basicity (greater pK_a_ value) of imine groups upon cross-linking of chitosan. The creation of unique imine surface adsorption sites through the modular synthesis reported herein is inferred to result in a modified bead surface upon cross-linking of chitosan, along with pillaring of the bead network to allow for the chelation of Ca^2+^ at the bead periphery [54].

The adsorption studies revealed a strong correlation of adsorption capacity with the surface adsorption sites and the textural properties of the beads. The attenuated diffusion within the bead core and limited swelling of the cross-linked beads provide support that surface adsorption sites play a key role. This finding correlates with the adsorption capacity and limited accessible surface area of the beads, as compared with adsorbents in a powdered form [64,65]. Restricted diffusion at the core domains of the bead is supported by the kinetics of sulfate adsorption for a model bead system (cf. Appendix A). In addition, the heterogeneity factor *n* of the bead materials concerning sulfate uptake increased (cf. Appendix A). This heterogeneity parameter (*n*) further supports that multiple adsorption sites contribute to the overall uptake of sulfate in this system. In conjunction with the kinetic results (cf. Appendix A), the presence of multiple adsorption sites is supported by the best-fit results provided by the pseudo-second order kinetic model. The offset in uptake between beads versus powders relates to the greater accessible surface area of powder materials, in agreement with the sulfate Q_m_ values for select adsorbent materials from the literature (cf. Table 2). While sodium sulfate has a favorable water solubility, sulfate displays lower affinity towards chitosan than other oxyanions such as vanadium (III) oxide (V_2_O_3_). The greater uptake of V_2_O_3_ may relate to its less negative Gibbs energy of hydration, in line with its lower water solubility (ca. 0.1 g/L) [66] versus sodium sulfate (ca. 140 g/L). In another study, the simultaneous uptake of Cu^2+^ and sulfate anions onto different forms of cellulose was noted, where significant uptake occurs as Cu^2+^ chelates with the electron-rich domains of cellulose. Thereafter, sulfate is co-adsorbed due to charge neutralization and complexation effects [67].

### 2.6. Bead Regeneration Studies

Previous reports indicate that biopolymer adsorbents may undergo loss of adsorption capacity after multiple cycles of adsorption-desorption. A key feature of sustainable adsorbent technology relates to a near constant level of removal efficiency after regeneration, along with minor susceptibility to degradation over time. Thus, the most promising bead material (5:1 calcium imbibed system) was evaluated for its sulfate adsorption-desorption properties. Sodium chloride solution is a viable agent for sulfate desorption, however, in case of the weakly coordinated Ca^2+^ species, competitive Na^+^ binding effects may lead to calcium leaching. To assess this effect, two Ca^2+^ imbibed bead systems (1:1 and 5:1 cross-linked systems) were tested using NaCl solution as the regenerant. Therefore, the supernatant was tested for Ca^2+^ leachate with Eriochrome Black T as the indicator [73]. Both the 5:1 and 1:1 systems showed some leaching, whereas the 1:1 system was less evident. To address leaching effects, the regeneration of beads used 0.1 M CaCl_2_ solution as an alternative regenerant media. The results for the 5:1 cross-linked bead material are presented in Figure 6. Prior to cycle 1 (cf. Figure 6), the regeneration studies required soaking of the beads for 24 h in Millipore water. The beads were then added to the sulfate solution (1000 ppm), where six cycles of regeneration were carried out. In general, a slight increase occurs from the first to the last cycle, indicating that these bead systems have good stability overall across multiple cycles of regeneration, according to the reported standard errors. A precedence for the regeneration method reported in Figure 6 finds support in a fixed-bed column study reported by Solgi et al. [74]. Therein, the recyclability of chitosan materials using aqueous CaCl_2_ revealed reproducible sulfate adsorption over multiple cycles of adsorption-desorption.

## 3. Materials and Methods 

### 3.1. Materials

CaCl_2_, BaCl_2_, FeCl_3_, glutaraldehyde (GA), EDTA, low-molecular-weight chitosan (75−85% deacetylation with a molecular weight range of 50,000−190,000 Da), KBr (IR Grade), and NaCl (ACS grade) were obtained from Sigma Aldrich Canada (Oakville, Ontario, Canada). Glacial acetic acid, HCl (*aq*) 37%, NaOH, and Na_2_SO_4_ were obtained from Fisher Chemical (Fisher Scientific, Geel, Belgium). Eriochrome Black T and *p*-nitrophenol (PNP) were purchased from Alfa Aesar (Alfa Aesar, Tewksbury, MA, USA), anhydrous sodium carbonate and NH_4_Cl were purchased from EMD (by Merck, Darmstadt, Germany). All chemicals were used as received unless specified otherwise. Millipore water (18.2 MOhm cm^−1^ Resistance) was used for the preparation of all solutions.

#### 3.1.1. Synthesis of Cross-Linked Chitosan Beads

Briefly, chitosan powder (5 g) was dissolved at 22 °C overnight in 250 mL 2% (v/v) acetic acid solution and then added dropwise using a volumetric burette with gentle stirring into a 0.5 M NaOH solution (250 mL). After complete addition, the stirring was stopped, and the beads material were left to neutralize under quiescent conditions for at least 16 h at 22 °C to afford stabilization of the beads after the phase-inversion process. The beads were rinsed with deionized water until the washings reached a neutral pH. The required amount of glutaraldehyde was calculated based on the desired stoichiometric molar ratio and the requisite volume of glutaraldehyde solution was added to the beads at pH 7. Thus, a 5:1 ratio corresponds to 5 moles glutaraldehyde to 1 mol of glucosamine monomers of chitosan. Sufficient water was added to submerge the beads to carry out the reaction for 48 h at 23 °C. Thereafter, the beads were washed with copious amounts of water and imbibed in fresh water for 48 h with repeated exchange of water to allow efficient removal of unreacted glutaraldehyde. The beads were either dried at 40 °C for 48 h or used without drying for subsequent calcium-imbibing, followed by drying at 40 °C for 2 days.

#### 3.1.2. Calcium Doping

The beads were imbibed in excess volume of aqueous CaCl_2_ (0.1 M) for 48 h at 22 °C under quiescent conditions.

### 3.2. Methods

#### 3.2.1. Materials Characterization

The weight loss profiles were obtained using a Q50 TA Instruments (New Castle, DE, USA) thermogravimetric analyzer (TGA) with a weighing precision of 0.01% and a temperature precision of 0.1 °C for isothermal conditions. Samples were heated in open aluminum pans at 30 °C for 5 min to enable equilibration prior to heating at 5 °C/min to 500 °C. The PZC measurement is based on the pH shift method, where solutions (25 mL) at several different pH (ca. 2, 4, 5, 8, 10) values were used with the addition of chitosan beads (ca. 60 mg). After mixing for 48 h at 22 °C to allow for equilibration, the pH of each solution was re-measured (with a Mettler Toledo Seven Compact and accumet probe with accuracy of pH ± 0.2; Mettler Toledo, Mississauga, ON, Canada) and the PZC was determined by calculation of the intersection of the plotted lines of initial vs. final pH [75]. Solids ^13^C NMR spectra were obtained using a wide-bore (89 mm) 8.6 T Oxford superconducting magnet system with a 4 mm rotor that used cross-polarization conditions with magic angle spinning (CP-MAS) during acquisition. Topspin 1.3 (Bruker Bio Spin, Billerica, MA, USA) was used for the control of acquisition and data processing parameters. Determination of the sulfate concentration was carried out via UV-Vis-based turbidity spectrophotometric measurement that employs BaCl_2_ (Indian Standard IS 3025) with a Thermoscientific Spectronic 200e (Ottawa, ON, Canada). All X-ray Photoelectron Spectroscopy (XPS) measurements were acquired using a Kratos (Manchester, UK) AXIS Supra system. This system is equipped with a 500 mm Rowland circle monochromated Al K-α (1486.6 eV) source and a combined hemi-spherical analyzer (HSA) with a spherical mirror analyzer (SMA). A spot size of hybrid slot (300 × 700 microns) was used. All survey scan spectra were collected in the -5-1200 binding energy range in 1 eV steps with a pass energy of 160 eV. An accelerating voltage of 15 keV and an emission current of 15 mA was used for the analysis. Solvent swelling was estimated gravimetrically by fully soaking the bead materials in Millipore water for 24 h, where excess solvent was removed from the bead surface by tamping dry with tissue paper. The residual of bound water within the swollen beads was removed upon drying until no further weight loss was measured. The swelling capacity was calculated by the dry weight value (100%) vs. the wet weight after soaking. 

Surface area was estimated using *p*-nitrophenol (PNP), where a 5 mM PNP solution was prepared in 0.1 M carbonate buffer at pH 8.4 ± 0.2 to yield the anion form of PNP. The bead systems (ca. 15 mg) were added to 10 mL of 5 mM PNP solution at 22 °C and shaken for 24 h (Scilogex SK-0330-Pro, Scilogex, Rocky Hill, CT, USA). UV-vis absorbance measurements for concentration determination was carried out at the isosbestic point of the spectral profile of PNP to avoid pH-dependent intensity variations.

#### 3.2.2. Adsorption Studies

Approximately 10 mg of adsorbent was added to a stock sulfate solution (10 mL) with a known concentration (ca. 50 to 1000 ppm). The pH of the solution was adjusted to pH 5 with hydrochloric acid solution. The samples were equilibrated at 22 °C for 24 h with constant shaking (Scilogex SK-0330-Pro). The difference in concentration (C_0_-C_e_) before and after the adsorption process was used to calculate Q_e_ (cf. Equation (2)). The sulfate concentration was determined via turbidity measurements (Indian Standard IS 3025 Part 24, 4, re-affirmed 2009) with BaCl_2_ and conditioning reagent (2) at 420 nm. The Sips model (sometimes referred to as Langmuir–Freundlich isotherm) was used to evaluate the isotherm profile and, depending on the exponent term (*1*/*n*), it may converge either to the Freundlich (approaching zero) or to the Langmuir model (approaching unity) [60].
(2)$$Q_{e}=\frac{Q_{m}a_{s}C_{e}^{1/n}}{1+a_{s}C_{e}{}^{1/n}}$$

The parameters in Equation (2) are defined as follows: Q_m_ is the monolayer adsorption capacity (mg/g), a_s_ is the Sips constant, and 1/*n* is the index heterogeneity of the sorbent material.

#### 3.2.3. Calcium Leaching

To test for calcium leaching in the sample solution, a sample solution (5–10 mL) was added to 25 mL water, 5 mL NH_4_Cl/NH_4_OH buffer (4 M, pH between 10.2–10.4) with a few drops of Eriochrome Black T indicator added. Titration with EDTA solution was done until a color change occurred at the end-point [73].

#### 3.2.4. Bead Regeneration Studies

Adsorption and regeneration cycles were carried out by pre-swelling the beads in Millipore water to ensure equilibrium conditions and eliminate inconsistent swelling behavior. The respective beads were transferred into the appropriate solutions (1000 ppm sulfate solution for adsorption; 0.1 M CaCl_2_ solution for bead regeneration) after washing and quick removal of excess water by tamping dry with filter paper. 

#### 3.2.5. Bead Size Determination

Bead size determination was estimated using a digital caliper with a bead sample size of *n* = 15 per system.

## 4. Conclusions

Herein, several types of chitosan bead systems were prepared using a modular synthetic design strategy (cf. Figure 1) that involves phase-inversion synthesis along with glutaraldehyde cross-linking at variable levels with sequential surface modification via bead doping with Ca^2+^ in aqueous calcium chloride. Complementary characterization of the bead systems was carried out using ^13^C NMR spectroscopy, XPS, and thermal analysis to provide complementary support of variable cross-linking and Ca^2+^ imbibing. Cross-linking of chitosan beads was achieved using excess stoichiometric levels of glutaraldehyde, where the cross-linker content correlates with the solvent swelling and dye adsorption properties of the biopolymer network. The sulfate adsorption isotherm results revealed that greater uptake occurred upon incremental cross-linking, along with Ca^2+^ imbibition. This approach finds support in other work, albeit with other metal-oxyanion (CrO_4_^2−^) systems and with variable types of co-adsorbed cation species [35,43,44,45,46]. The sustainability of the chitosan-based adsorbent materials prepared herein was demonstrated with the use of a modular green design and facile approach. Recyclability of beads was shown using a salt-based regenerant (CaCl_2_) solution over multiple cycles of adsorption-desorption to mitigate potential leaching effects of calcium from the bead system. Further work is underway to scale-up the process for water treatment applications that will focus on the controlled removal of oxyanions under dynamic conditions relevant to food, pharmaceuticals, and nutraceuticals applications [76]. 

## Figures and Tables

**Figure 1 ijms-21-07130-f001:**
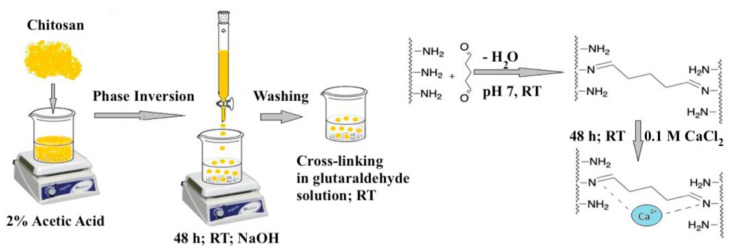
Schematic view of the synthesis of chitosan-derived beads under phase inversion synthesis with subsequent cross-linking via Schiff-base formation with glutaraldehyde and calcium-ion imbibing.

**Figure 2 ijms-21-07130-f002:**
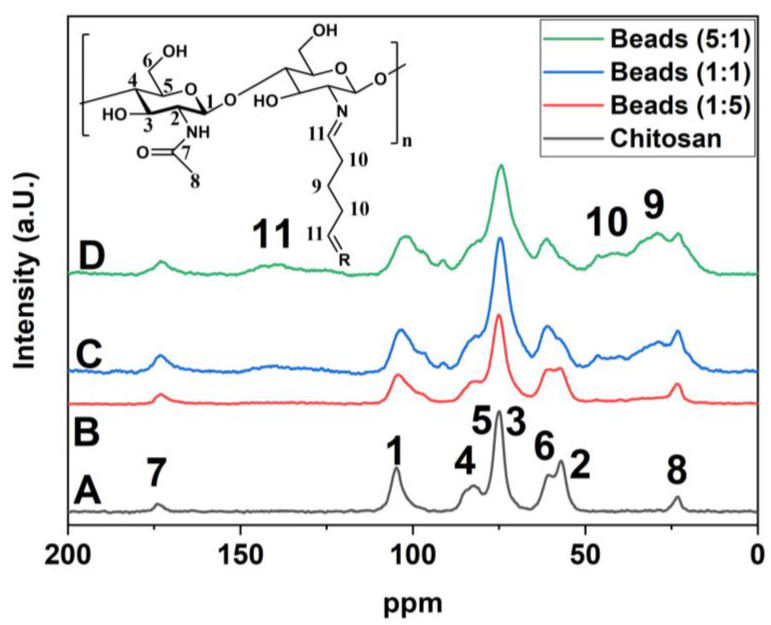
Solids ^13^C CP-MAS NMR spectra at 125 MHz: chitosan (no calcium) (A), calcium imbibed beads cross-linked at variable cross-linking (CL) ratios of glutaraldehyde to chitosan monomer units of 1:5 (B), 1:1 (C) and 5:1 (D).

**Figure 3 ijms-21-07130-f003:**
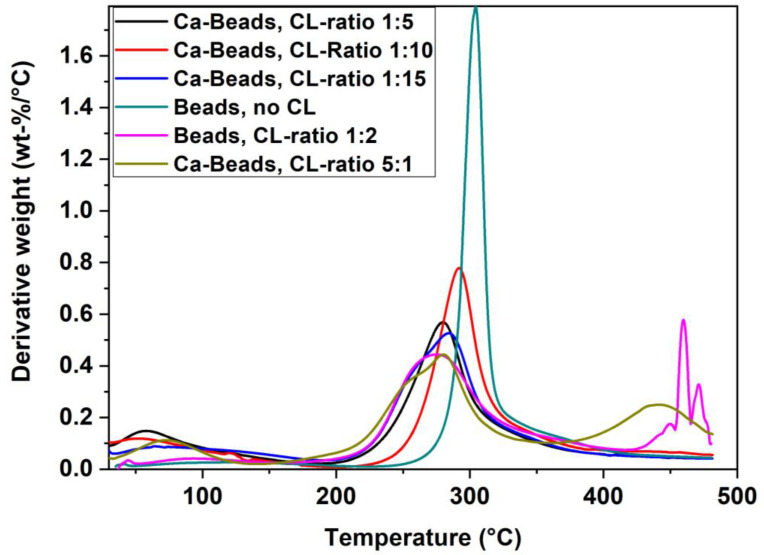
TGA profiles of the calcium doped beads with variable cross-linking (CL) ratios relative to the non-imbibed chitosan beads without cross-linking (no CL). The derivative weight scaling is truncated for the latter bead system.

**Figure 4 ijms-21-07130-f004:**
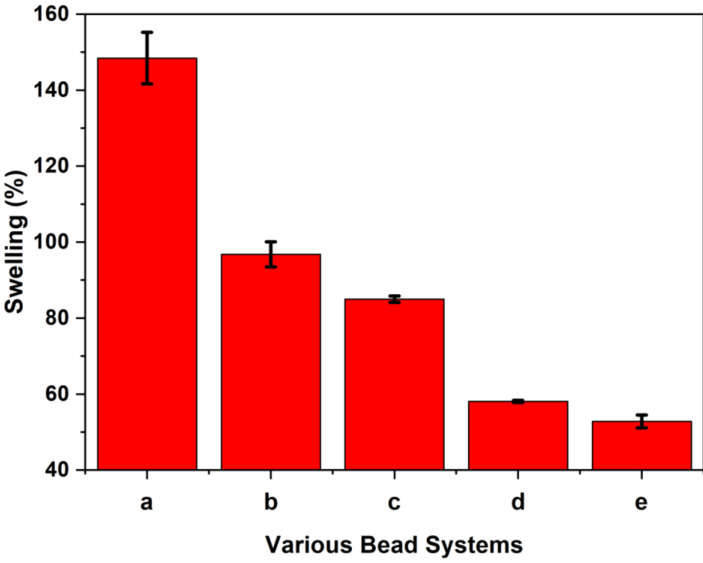
Swelling (%) for chitosan bead systems: non-modified beads (a), calcium imbibed beads with a CL- ratio of 1:10 (b), non-imbibed beads with a CL-ratio of 1:5 (c), calcium imbibed beads with a CL-ratio of 1:1 (d) and 5:1 (e).

**Figure 5 ijms-21-07130-f005:**
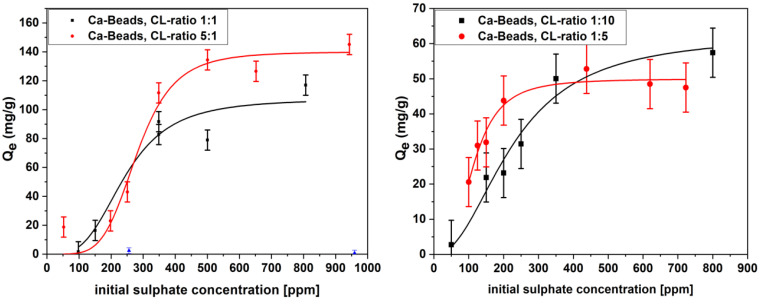
Sulfate adsorption isotherms with cross-linked and calcium imbibed beads (CL-ratio of 1:1 and 5:1 on the left, CL-ratio of 1:5 and 1:10 on the right (for beads with CL-ratio 1:15, see Appendix A) at pH 5 and 295 K. The solid lines are the best-fit according to the Sips isotherm model (cf. Equation (2)).

**Figure 6 ijms-21-07130-f006:**
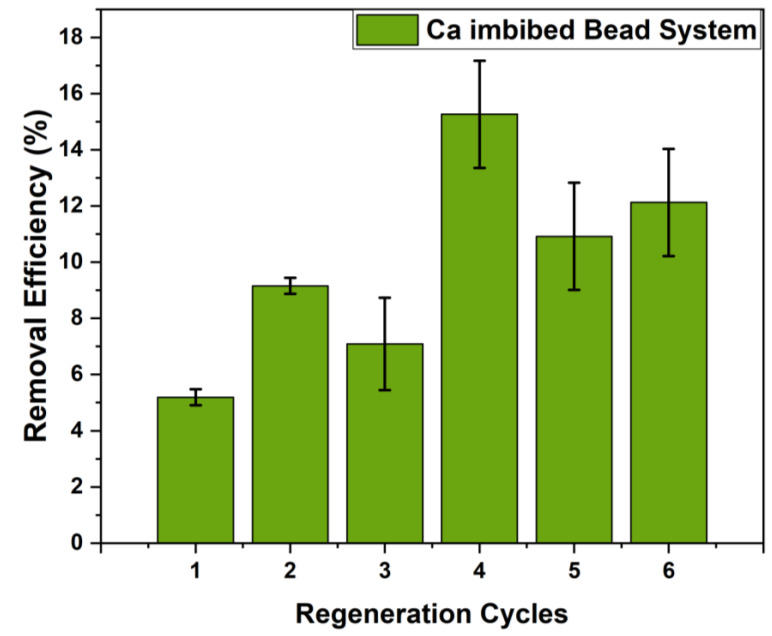
Removal efficiency of 5:1 cross-linked, calcium-imbibed bead systems with ca. 1000 ppm sulfate solution at pH 4–5 over 6 adsorption/desorption cycles and regeneration in 0.1 M CaCl_2_-solution.

**Table 1 ijms-21-07130-t001:** Dye-based Surface Area (SA) using the anion form of PNP at pH 8.4 ± 0.2.

Bead System	SA (m^2^g^−1^)
5:1 w/ Ca^2+^	182 ± 19
1:1 w/ Ca^2+^	153 ± 4
1:10 w/ Ca^2+^	75 ± 32
unmodified	110 ± 72

The molar surface area of PNP relates to its orthogonal orientation (25 × 10^−20^ m^2^) where Y = 1 (see Equation (1)), and the absorbance was measured at the isosbestic point of PNP (λ = 347 nm).

**Table 2 ijms-21-07130-t002:** Literature comparison of the sulfate monolayer uptake capacity (Q_m_) values for different sorbent materials and results from this study.

Material	SO_4_^2−^ Uptake Q_m_ (mg/g)	Concentration (ppm)	Conditions
*Cross-linked* *chitosan beads* (This study) *	35	C_0_ = 1000 ppm	CL-ratio 1:15 Ca^2+^-imbibed pH 5
*Cross-linked* *chitosan beads* (This study) *	140	C_0_ = 1000 ppm	CL-ratio 5:1 Ca^2+^-imbibed pH 5
ZnCl_2_ modified zeolite [68]	38	C_0_ = 1000 ppm	CTAB modified 40 °C
Pyrrole modified activated carbon [69]	45	C_0_ = 250 ppm	pH 7
Modified Rice Straw [70]	75	C_0_ = 500 ppm	pH 6.4
Poly(*m*-phenylendiamine) [71]	109		pH < 3
Chitin [64]	150	C_0_ = 1200 ppm	pH 4.3 10 g/L sorbent
Chitosan [72]	210	C_0_ = 2500 ppm	Fe(II)SO_4_;
CC/QAC [67]	526		pH 5; CuSO_4_

CC/QAC refers to Carboxylate Cellulose/Quarternised Cellulose. * Denotes results obtained in this study

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
