# Peer review of "Modular Chitosan-Based Adsorbents for Tunable Uptake of Sulfate from Water"

_ijms, 2020, doi:10.3390/ijms21197130_

Round 1
Reviewer 1 Report
The manuscript is interesting, however, there are some suggestions for authors:
- In the part of the introduction, it is not clear for the reader to read the manuscript. Please, re-write this part.
- In the research, Can chitosan dissolve with other material stuff together to be beads?
- How much molecular weight of chitosan to use for the study?
- The authors mentioned that the results of this study are relevant to diverse pharmaceutical and nutraceutical applications, however, I did not read about the information about it. What kind of pharmaceutical and nutraceutical applications?
- Can the authors provide the other experimental data or information to prove chitosan beads with higher cross-inking? such as Transmission Electron Microscopy?
Author Response
Reviewer 1:
- In the part of the introduction, it is not clear for the reader to read the manuscript. Please, re-write this part.
Author Response: The introduction was modified accordingly, as recommended by the reviewer.
- In the research, Can chitosan dissolve with other material stuff together to be beads?
Author Response: In its dissolved form, other biopolymers (e.g. alginate) or metal salts or composites can be blended with chitosan (see doi link below) in aqueous solution to form composites with variable solubility. Fig. 1 in the manuscript shows the phase-inversion synthesis of chitosan beads where the resulting composite is insoluble in aqueous media, especially at pH values above the pKa of chitosan.
http://doi.org/10.1016/j.ijbiomac.2019.02.059
- How much molecular weight of chitosan to use for the study?
Author Response: The molecular weight of chitosan used was 50,000-190,000 Da with 75-85% DA, obtained from Sigma-Aldrich, as mentioned in Materials and Methods.
- The authors mentioned that the results of this study are relevant to diverse pharmaceutical and nutraceutical applications, however, I did not read about the information about it. What kind of pharmaceutical and nutraceutical applications?
Author Response: References are cited in the bibliography that outline the use of chitosan in various pharmaceutical and nutraceutical technological applications (cf. Mar. Drugs 2015, 13, 1819-1846). For example, chitosan was used for insulin formulations as a treatment strategy in diabetes therapy (cf. Colloids Surf. B Biointerfaces 2006, 53, 193–202). Similarly, the development of orally ingestible adducts for controlled uptake of lipids or anionic waterborne contaminants such as dextran sulfate or controlled-release of chondroitin sulfate represents a key strategy for the treatment of osteoarthritis (cf. Current Drug Delivery, 2018, 15(6), 907-916). The reviewer is referred to refs. [47-49, 77] in the revised manuscript.
- Can the authors provide the other experimental data or information to prove chitosan beads with higher cross-inking? such as Transmission Electron Microscopy?
Author Response: Despite the role of microscopy for characterizing morphology and other surface features, the relevance of TEM is limited for determination of cross-linking. Instead, we have established the role of cross-linking by NMR, and other complementary techniques such as TGA, as described in the manuscript. Specifically, the presence of imine-linkages are evidenced by 13C-NMR where the appearance of the characteristic signals (ca. 140 ppm) for the imine signature. As well, further indirect support for different cross-linking ratios can be inferred from the swelling data as well as shift in PZC from 6.5 to 6.8 (whereas chitosan has a pKa ~ 6.3) with increase in cross-linking.
In closing, we acknowledge the constructive and insightful comments of Reviewer #1. The authors appreciate the opportunity to improve the quality of the manuscript submission. We have further edited the manuscript for language, syntax, and clarity throughout to meet the high standards of this journal.
Reviewer 2 Report
The present work on "Modular Chitosan-Based Adsorbents for Tunable Uptake of Sulfate from Water" by Steiger et al described a modular design of chitosan beads with tunable sulfate adsorption properties via a phase-inversion synthesis which can offer controlled sulfate removal from contaminated water as well as promising tool for controlled adsorption-desorption of inorganic and organic sulfate from a diverse range of environmental and pharmaceutical.
I found the manuscript to be well-written. The introduction section was providing sufficient background on why such a research was needed and how this approach can serve as a promising tool for several applications.
In regards to the research design, I highly suggest authors to include additional characterization of their beads synthesis process including particle size, distribution, PDI, surface charge, etc using systems like particle size analyzer to further show reproducibility and validate the uniformity of this system. Additionally, including microscopic evaluation of these bead after synthesis as well as before and after sulfate adsorption can provide more visual information to better understand these results.
For Figure 3 on Swelling (%) for chitosan bead systems with various CL-ratios and Figure 5 on Removal efficiency of cross-linked, calcium-imbibed bead systems with sulfate solution , I highly recommend authors to perform appropriate statistical analysis to show and describe how significant their findings are and help readers gaining more meaningful insight from described results.
Author Response
Reviewer 2:
The present work on "Modular Chitosan-Based Adsorbents for Tunable Uptake of Sulfate from Water" by Steiger et al described a modular design of chitosan beads with tunable sulfate adsorption properties via a phase-inversion synthesis which can offer controlled sulfate removal from contaminated water as well as promising tool for controlled adsorption-desorption of inorganic and organic sulfate from a diverse range of environmental and pharmaceutical.
I found the manuscript to be well-written. The introduction section was providing sufficient background on why such a research was needed and how this approach can serve as a promising tool for several applications.
- In regards to the research design, I highly suggest authors to include additional characterization of their beads synthesis process including particle size, distribution, PDI, surface charge, etc using systems like particle size analyzer to further show reproducibility and validate the uniformity of this system. Additionally, including microscopic evaluation of these bead after synthesis as well as before and after sulfate adsorption can provide more visual information to better understand these results.
Author Response Given the large bead size (~ 1-2 mm diameter) in the dry state, the PDI estimate is not a suitable method. Instead, we include the size distribution (standard deviation, mean, median and spread) of the values measured macroscopically with a digital caliper. Since all adsorption was measured at pH < PZC, the bead would possess a positive surface charge, in line with the pKa for chitosan (6.3) since the measured PZC of the beads was ~ 6.5-6.8.
The surface area determination was performed via a dye probe method in aqueous media. However, we provided a graphical image for better visualisation of the beads. As well, the size distribution (mean, median, standard deviation and spread) is seen in Figure S8 and reported numerically in Table S2.
Figure S8: Size of different bead systems from left to right: non-modified, 1:15 Cl-ratio, 1:10 Cl-ratio, 1:5 Cl-ratio, 1:1 Cl-ratio and 5:1 Cl-ratio.
Table S2: Bead size estimates* in the dry state.
|
Bead System |
Mean (mm) |
Median (mm) |
Standard Dev. (mm) |
Spread (mm) |
|
Non-modified |
1.51 |
1.46 |
0.17 |
0.60 |
|
1:15 Cl-ratio |
1.34 |
1.32 |
0.20 |
0.81 |
|
1:10 Cl-ratio |
1.28 |
1.29 |
0.18 |
0.64 |
|
1:5 Cl-ratio |
0.96 |
0.93 |
0.14 |
0.50 |
|
1:1 Cl-ratio |
1.42 |
1.43 |
0.15 |
0.56 |
|
5:1 Cl-ratio |
1.48 |
1.49 |
0.17 |
0.66 |
* The mean size was obtained using a digital caliper using 15 individual measurements
The results show variation that parallel the synthetic modification of non-cross-linked vs. cross-linked systems in agreement with Mahninia et al. (see Ref. 26). Spectroscopic evaluation of the beads in the dry state does not provide meaning information due to the role of swelling. Our attempts to image the beads in the wet state before and after adsorption was not feasible due to background fluorescent interference. Instead, we studied the regeneration of beads and demonstrate that the structure was stable over multiple cycles of adsorption-desorption.
- For Figure 3 on Swelling (%) for chitosan bead systems with various CL-ratios and Figure 5 on Removal efficiency of cross-linked, calcium-imbibed bead systems with sulfate solution , I highly recommend authors to perform appropriate statistical analysis to show and describe how significant their findings are and help readers gaining more meaningful insight from described results.
Author Response: Fig. 3 provides two types of information; the temperature range and the mass loss associated with the thermal event. The data in fig. 3 are based on a 1st derivative of the mass loss (%) vs. temperature [weighing precision 0.01%; isothermal temperature precision 0.1 °C]. From the viewpoint of materials characterization, we evaluate the temperature (maxima) and the full-width-half-maximum to give a qualitative estimate of the thermal stability of the material. This is often satisfactory for distinguishing structural variations that can be related to thermal stability. Given the qualitative manner in which TGA is discussed herein, the use of statistics would represent an over-interpretation in light of the discussion for Fig. 3. We have interpreted our results accordingly to emphasize trends in relation to the modular synthesis described in fig. 1. This approach is in line with a large body of research using this TGA method.
In closing, we acknowledge the constructive and insightful comments provided by reviewer #2. The authors appreciate the opportunity to improve the quality of this manuscript submission. We have further edited the manuscript for language, syntax, and clarity throughout to meet the high standards of this journal.

Round 2
Reviewer 1 Report
No comments or suggestions, after authors modified the manuscript. Thanks.